# Trends in Diagnosing Obstructive Sleep Apnea in Pediatrics

**DOI:** 10.3390/children9030306

**Published:** 2022-02-24

**Authors:** Mandip Kang, Fan Mo, Manisha Witmans, Vicente Santiago, Mary Anne Tablizo

**Affiliations:** 1Department of Medicine, University of California San Francisco-Fresno, Fresno, CA 93701, USA; fan.mo@ucsf.edu (F.M.); mtablizomd@gmail.com (M.A.T.); 2Department of Pediatrics, Faculty of Medicine & Dentistry, University of Alberta, Edmonton, AB T6G 2R3, Canada; manishawitmans@gmail.com; 3Kaiser Permanente Health Care System, Frenso, CA 93726, USA; vsantiagomd@gmail.com; 4Department of Pediatrics, Stanford University, Palo Alto, CA 94304, USA; 5Department of Pediatrics, Valley Children’s Hospital, Madera, CA 93720, USA

**Keywords:** pediatrics, obstructive sleep apnea, diagnostics, polysomnography

## Abstract

Obstructive sleep apnea in children has been linked with behavioral and neurocognitive problems, impaired growth, cardiovascular morbidity, and metabolic consequences. Diagnosing children at a young age can potentially prevent significant morbidity associated with OSA. Despite the importance of taking a comprehensive sleep history and performing thorough physical examination to screen for signs and symptoms of OSA, these findings alone are inadequate for definitively diagnosing OSA. In-laboratory polysomnography (PSG) remains the gold standard of diagnosing pediatric OSA. However, there are limitations related to the attended in-lab polysomnography, such as limited access to a sleep center, the specialized training involved in studying children, the laborious nature of the test and social/economic barriers, which can delay diagnosis and treatment. There has been increasing research about utilizing alternative methods of diagnosis of OSA in children including home sleep testing, especially with the emergence of wearable technology. In this article, we aim to look at the presentation, physical exam, screening questionnaires and current different modalities used to aid in the diagnosis of OSA in children.

## 1. Background

Although the prevalence of obstructive sleep apnea (OSA) is difficult to obtain in the pediatric population, the current literature reports rates of about 0.7% to 13% [1]. The wide range can be attributed to the various thresholds of defining OSA in children and methods used in the diagnosis of OSA, including the use of hemoglobin oxygen saturation (SpO2) alone, SpO2 including airflow, and SpO2 including airflow and respiratory effort. Only a handful of studies have used polysomnography (PSG) to help better quantify prevalence, and even in those reports the subject populations were limited (*n* = 12–50) [2,3,4,5,6,7,8,9]. Nonetheless, however wide the distribution of prevalence may be, it has been well established that OSA, a form of Sleep Related Breathing Disorder (SRBD), can lead to gas exchange abnormalities along with fragmented and insufficient sleep. There is evidence in the literature that untreated OSA in children is associated with behavioral and neurocognitive problems, impaired growth and in the long run can lead to cardiovascular and metabolic consequences. Recent newborn data show that insufficient sleep can lead to obesity and obesity-related illnesses [10]. Setting aside the health risks associated with OSA in pediatrics, the social consequences can also be very burdensome with disturbances in familial, education, and psychological development of the child. Given the physical and psychological detrimental effects of OSA, it is becoming more apparent that early and adequate diagnosis is imperative in the hopes of preventing long term sequelae.

## 2. Objective

Our objective is to highlight current methods of diagnosing OSA in the pediatric population. We also aim to highlight and compare innovative alternative methods of testing for OSA besides the conventional PSG.

## 3. Screening for OSA

### 3.1. Pediatric Sleep History

A comprehensive sleep history is the essential first step towards the diagnosis of OSA in children. A detailed and thorough sleep history will provide a foundation of diagnosing sleep disorders including but not limited to OSA, and possibly any comorbid disorders such as insomnia, restless leg syndrome, etc. Depending on the age of the patient, clinicians should obtain clinical information from parents, other family members and/or caretakers. Older children should be asked about their own perception of symptoms much like adults. The prevalence rate of habitual snoring is up to 27.6% [11]. Even though snoring may be the most common presenting symptom of SDB, the generally reported prevalence of OSA is only about 1–5% [11]. The quality of snoring does not predict the presence and severity of OSA [11]. In addition to habitual snoring and abnormal breathing patterns (choking, gasping and apnea), clinicians should also focus on sleeping positions such as hyperextended neck, secondary nocturnal enuresis, behavioral disorders described as inattentiveness, poor mood regulation, irritability, and hyperactivity during the day. In younger infants, snoring may not be as prominent. Freezer et al. revealed that 53% of infants with the diagnosis of OSA presented with failure to thrive [12]. The pediatric population presents differently compared to adults when sleep quality is disturbed and is less likely to present with daytime sleepiness revealed by Gozal et al. In this paper, only 7 out of 54 pediatric OSA patients presented with sleepiness [13]. Pediatric patients with OSA are more likely to present with inability to concentrate, depressed mood, and behavioral issues especially in younger ages [14,15]. Sleep logs or diaries can also be useful in providing sleep wake schedules, feeding time for infants and toddlers, nap times, and total sleep time to be able to differentiate sequelae of OSA from insufficient sleep, poor sleep hygiene, sleep behavioral insomnia of childhood and delayed sleep circadian rhythm disorder in teenagers. Thorough history will help in screening possible OSA but the positive and predictive value of history alone in the diagnosis of OSA is only 65% [11].

### 3.2. Pediatric Sleep Questionnaires

To help with early recognition of any sleep-related disorders, researchers have developed various screening tools to help clinicians obtain a more focused sleep history. The focus of the sleep history varies based on the pediatric patient’s age. During infancy, a brief infant sleep questionnaire (BISQ) designed by Saheh is a tool for screening sleep history for the age group 0–29 months of age [16]. The BISQ questionnaire evaluates several categories including nocturnal sleep duration, night awakenings, method of falling asleep and parent behavior. Many other available questionnaires focus on sleep history for older children and adolescent groups [17]. One of the most utilized questionnaires was developed by Chervin et al., the pediatric sleep questionnaire (PSQ) [14]. To identify the pediatric population with sleep-related breathing disorders, Chervin et al. developed a pediatric sleep questionnaire that consisted of a total of 22 questions to be used among clinicians. The questionnaire includes three main domains including assessment for snoring, sleepiness along with OSA related symptoms, and inattentive/hyperactive behavior categories [14]. The age range for this questionnaire is from 2 years to 18 years old. The sensitivity and specificity of the PSQ was noted to be 0.87 and 0.85, respectively, when 8 or more of the questions were answered “yes” from the 22 questions [14]. Kadmon et al. also developed a shorter version of questionnaire with eight questions termed I’M SLEEPY, which serves as a screening tool for pediatric OSA with sensitivity and specificity of 82% and 50%, respectively [18]. Daytime sleepiness is often commonly present among adult patients with SDB and the Epworth sleepiness scale has been developed and utilized among clinicians and researchers for adult populations [19]. A modified Epworth sleepiness scale has been designed to screen any daytime sleepiness among children with questions that are more appropriate for pediatric populations [20]. The modified Epworth sleepiness scale for children changed two situations where sleepiness was assessed more specifically for children: sleepiness after lunch and while doing homework or taking a test.

In 2021, a systematic review and meta-analysis was conducted which evaluated the precision of pediatric screening questionnaires for OSA and compared them with gold standard polysomnography for those less than the age of 18 years. In the review, 13 studies and two questionnaires including the sleep-related breathing disorder scale of the pediatric sleep question, aka the SRBD-PSQ, and OSA-18 met the criteria for quantitative synthesis. The review showed that the SRDBD-PSQ had a slightly higher sensitivity (0.76 vs. 0.56), however the OSA-18 had a higher specificity than the SRBD-PSQ (0.73 vs. 0.43) [21]. These findings suggest that even the best tools that are available are far from perfect and do not adequately capture children with OSA. In addition, these screening questionnaires also do not include the composite of parameters that would also influence the likelihood of identifying OSA such as the history of prematurity, asthma, GERD, etc. Based on these, the sleep questionnaires are used as screening tools and is insufficient to make the diagnosis of OSA in children [11].

### 3.3. Pediatric Sleep Physical Examination

In general, when conducting a physical examination, a lot of information can be attained with just simple observation of the child. A child’s level of alertness can be assessed by observing any of the following: repetitive yawning, daytime sleepiness, frequent changes in position, overactivity, inattention, irritability, and age-appropriate emotional dysregulation, all indicating possible lack of sleep [22,23,24]. These features suggest that the child’s sleep quality or sleep quantity may not be restorative. Growth/weight charts plotted on growth curves are important to document extremes such as poor growth or obesity, which can both be associated with severe OSA [11,25]. Following general observations and vital signs, it is imperative to do a thorough head and neck examination. Craniofacial anomalies such as midface hypoplasia, retrognathia, and micrognathia may suggest upper airway obstruction. Long (adenoidal) facies, mouth breathing, hypo nasal speech, and nasal congestion with decreased nasal airflow can be consistent with adenoidal hypertrophy which can contribute and/or cause OSA [22,23,24]. Additional features of nasal obstruction or congestion, including atopic features, may include dark circles under the eyes, swollen eyes, or mucosal or turbinate swelling which cannot distinguish between allergic versus non-allergic rhinitis. Additional clues such as a transverse nasal crease across the bridge of the nose, allergic salute, and Dennie Morgan lines may suggest atopic disease. Mallampatti scores can be used in older children. Tonsils are scored according to size on a scale from Grade 0–4 during the awake examination with grade 4 as the most severe with bilateral tonsils extending to midline and almost in contact with each other.

OSA in children can be due to multiple different pathologies of the airway including abnormal maxillomandibular development (with noted features being high-arched and narrow hard palate, crossbite, overlapping incisors or overbite), reduced flow through the upper airway (nasal stenosis), poor pharyngeal/laryngeal tone (due to neuromotor disease i.e., cerebral palsy or muscular dystrophy) or oropharyngeal crowding (looking at tonsillar grade), macroglossia, and high Mallampati score [22,23,24]. It is noteworthy to mention that tonsillar grade does not have a linear correlation to the diagnosis of OSA. The size of the tonsils also does not correlate with the severity of OSA. Since the pathophysiological process involved in OSA is the dynamic, sleep-related airway obstruction, it stands to reason that any awake assessments may not be sensitive or specific enough to determine the nature and severity of the airflow obstruction during sleep. According to one literature, the positive and predictive value of using physical examination alone is as low as 46% [11].

## 4. Diagnostic Testing

### 4.1. Levels of Testing

There are four different study types that can be conducted to test for OSA. They are classified by four levels. Level I is a polysomnography which has minimum of seven parameters: Electrooculography (EOG), Electroencephalography (EEG), chin electromyography (EMG), airflow, respiratory effort, oxygen saturations and electrocardiography (ECG) and is attended by a sleep technician. Level II also has a minimum of seven parameters (same as previously mentioned) but is not attended by a sleep technician. Level III has a minimum of four parameters which include ECG/pulse, oxygenation saturations, two channels of respiratory effort, or one respiratory effort and one airflow channel which is also not attended by a sleep technician. Lastly, a level IV study is that which must include a minimum of three channels, of which one is airflow or include actigraphy, oxygen desaturation, and peripheral arterial tone. Level 4 can be attended or unattended by a sleep technician [26].

### 4.2. Polysomnography (Level I)

The gold standard for the diagnosis of OSA in pediatrics is attended in-laboratory PSG [16].

According to AASM guideline, attended PSG is indicated for various respiratory indications not limited to the following: (1) clinical suspicion of obstructive sleep apnea syndrome, (2) following adenotonsillectomy to assess any residual obstructive sleep apnea syndrome with preoperative evidence for moderate to severe obstructive sleep apnea, (3) for positive airway pressure titration in children, (4) clinical assessment suggestive of congenital central alveolar hypoventilation syndrome, (5) in children being considered for adenotonsillectomy to treat obstructive sleep apnea syndrome, (6) pressure requirement assessment if a child’s growth and development has occurred or if symptoms recur while on PAP therapy [27]. 

The American Academy of Pediatrics has also set out guidelines that recommends in-laboratory polysomnography (grade A) for children with snoring on a regular basis (≥3 nights/week) and who have any of the following complaints: labored breathing during sleep, gasps/snorting, observed apneas, nocturnal secondary enuresis, unusual sleep position, cyanosis, headache on awakening, daytime sleepiness, attention/hyperactivity, and learning problems, along with PE findings associated with OSA [11].

For the diagnosis of OSA in children through polysomnography, the following data are collected: electroencephalogram (bilateral central, frontal, occipital EEG), left and right electrooculogram (EOG), chin and leg electromyogram (EMG), airflow signals (oronasal thermal airflow sensor and nasal cannula pressure transducer), dual thoracoabdominal respiratory effort signals (plethysmography), oxygen saturation, body position, electrocardiogram, carbon dioxide monitoring for detection of hypoventilation (either by end tidal C02 or by transcutaneous C02 as surrogates of arterial pC02), and a synchronized PSG Video. Monitoring snoring is optional [28].

Performing a PSG in children requires specialized training in pediatrics to help children and their families adjust to the intensity of the monitoring. Polysomnography is generally conducted in a similar way as adults. The most obvious difference is in the position and placement of the leads in children because of their size compared to adults. There are also differences in the sleep staging, respiratory scoring, and assessment of severity in adults and children.

Differences in sleep stages for scoring sleep studies are broadly differentiated in infants versus children and adults. Infant criteria are used for children under 2 months post term referred to as infants and those older 2 months post term to 18 years of age referred to as children. The reader is referred to AASM scoring manual for more detailed explanation of the staging and scoring. Respiratory events are also scored differently in children versus adults. The criteria for scoring respiratory events in pediatrics can be used in patients <18 years. However, clinicians can choose to score respiratory events in children ≥13 years using adult criteria [28]. The criteria for scoring different types of respiratory events (apnea and hypopnea) are based on the number of breaths, of at least two breaths in children compared to a timed duration of 10 s in adults. It is imperative, in children, to monitor transcutaneous or end tidal carbon dioxide to assess for obstructive hypoventilation which is not seen in healthy adults with OSA [28]. Furthermore, diagnostic criteria and thresholds for diagnosis of OSA are different for children compared to adults. In adults, apnea hypopnea index (AHI) of 5–15 events per hour is considered mild OSA, >15–30 is moderate, and above 30 events are considered severe. However, in the pediatric population, the threshold for diagnosis of OSA is lower compared to adults. The severity scale commonly used in pediatrics is as follows: AHI of >1 to <5 is considered mild OSA, AHI of ≥5 to <10 moderate OSA, and an AHI ≥10 is severe OSA.

### 4.3. Out of Center Sleep Testing or Home Sleep Testing

According to the guidelines from AASM, the gold standard of diagnosing OSA in the pediatric population is an in-laboratory, attended, polysomnography due to its accuracy in diagnosing respiratory disturbances and sleep architecture changes [29]. However, there have been challenges for the attended in-laboratory polysomnography from limited access to a sleep center, the laborious nature of the test, the specialized expertise required for diagnosing infants and younger children and social/economic barriers, which further delay the diagnosis and treatments. There has been increasing research on validation and utilization of home sleep testing on pediatric populations. However, due to lack of sufficient evidence, AASM published in their position statement in 2017 that does not support the use of home sleep testing for diagnosing sleep disordered breathing in pediatric populations [11,30,31].

The home sleep tests currently available utilize different sensors in detecting OSA. HST that utilize respiratory parameters include various sensors such as nasal pressure transducer, oximetry, chest and abdominal wall movement and snoring channels. HST that utilize cardiovascular sensors utilize pulse rate, heart rate variability, pulse transit time and peripheral arterial tonometry in detecting OSA. Some home sleep test are akin to PSGs because of the number of sensors used including EEG. There are limited studies focused on exploring the role of home sleep testing for children. Masoud et al. conducted their research on validating the portable monitor device MediByte (level III) among pediatric populations between age 7–17 years old. Their failure rate was about 17%, which was comparable to other studies conducted using portable home sleep tests [32]. The sensitivity and specificity of the utility of the MediByte for OSA in children is >93% for those with AHI of >10 (severe OSA). It might have a role in those with high pre-test probability with positive questionnaires. Unfortunately, mild OSA is far more common in children and therefore the device may fail to detect OSA in otherwise healthy children.

Another novel approach as a continuum to OSA diagnosis that uses a single sensor is cardiopulmonary coupling (CPC). It is able to report sleep quality and distinguish Rapid Eye Movement (REM) and Non-REM sleep based on cardiopulmonary coupling. In 47 children between the ages of 2–12 yrs, OSA diagnosis was compared using oximetry, PSG and cardiopulmonary coupling. The area under the curve for receiver operator characteristics (ROC) for diagnosing OSA using Apnea/Hypoventilation Index (AHI) obtained through CPC and PSG were 0.8214 and 0.8741 respectively.. Another study also found that the CPC parameters accurately reflected sleep fragmentation and OSA severity in 117 children, aged 7.96 ± 3.54 years with OSA. It is adequate for severe OSA but for very mild OSA the findings of the testing may warrant further evaluation [33,34].

One study using watch peripheral arterial tonometry (WATCHPAT) compared with polysomnography (PSG) for the diagnosis of pediatric obstructive sleep apnea (OSA) was done in 36 patients between 8–15 years. The study was prospective, with patients simultaneously wearing watch PAT while undergoing PSG. There is excellent agreement between methods for the AHI (intraclass correlation coefficient [ICC]: 0.89; 95% confidence interval [CI], 0.40–0.96; *p* < 0.001) and oxygen desaturation index ODI (ICC: 0.87; 95% CI, 0.69–0.94; *p* < 0.001) with higher specificity on patients with severe OSA (91.3% specificity). Correlation between methods was very good for the ODI (r = 0.83; 95% CI, 0.67–0.90; *p* < 0.001) and moderate for the AHI (r = 0.64; 95% CI, 0.30–0.85; *p* < 0.001) [21]. In another study, a total of 38 adolescents (mean age 15.1 ± 1.4 years) with suspected OSA were assessed with the WP200 and standard PSG simultaneously. This study also showed high correlations and good agreements in AHI and mSaO2 between the WP200 and PSG [35]. There were also high concordances in AHI severity between the WP200 and PSG [35]. Further studies with larger samples and age groups are needed to determine correlation between two diagnostic methods.

In addition to advancing technology about portable medical devices to diagnose SRBD (sleep-related breathing disorder), there has also been development of devices that help investigate and analyze characteristics of children that will likely have spontaneous resolution of obstructive sleep apnea [36].

A systematic review was done in 2021 analyzing the diagnostic performance of portable monitors (PM) compared with PSG for the diagnosis of pediatric OSA. There were 20 studies analyzed with 7062 participants. The AHI measured by PSG compared to different PM showed a pooled sensitivity of 74% (95% CI: 66–80%) and pooled specificity of 90% (95% CI: 85–94%). This study showed the potential of PMs for screening pediatric OSA patients and may provide alternative method of diagnosis [37]. For clinicians that cannot get access to PSG, portable monitor utilizing oximetry can be a useful adjunct to assess for possible OSA. It has very little information, however, and cannot differentiate between different forms of SRBD.

A meta-analysis was conducted to compare the pooled sensitivity and specificity of the Pediatric Sleep Questionnaire (PSQ), Obstructive Sleep Apnea Questionnaire (OSA-18), and pulse oximetry (PO) according to OSAS severity. The PSQ exhibited the highest sensitivity (74%) for detecting symptoms of mild pediatric OSAS. The PSQ and PO had comparable sensitivity in screening moderate and severe pediatric OSAS (0.82 and 0.89 vs. 0.83 and 0.83, respectively). PO yielded superior specificity in detecting mild, moderate, and severe pediatric OSAS (86%, 75%, and 83%, respectively) than the PSQ and OSA-18 (all *p* < 0.05). Age, percentage of girls, index test criteria, methodology quality, and sample size significantly moderated sensitivity and specificity. For early detection of pediatric OSAS, the combined use of PSQ and PO is recommended when polysomnography is not available [37].

## 5. Additional Diagnostic Testing

### 5.1. Upper Airway Imaging

Imaging has been proposed as a possible screening method for OSA in the pediatric population, however there are only few studies with a small number of patients. One study looked at using lateral radiographs, comparing it to PSGs to diagnose sleep apnea [38]. It was a retrospective study of 19 children (15 male and 4 female) from the ages of 21 months to 11 years old that had lateral cervical radiographs taken followed by a PSG. A child was considered to have a positive radiograph if tonsillar/adenoidal hypertrophy was present. Results showed that out of the 10 patients with positive radiographs, only 8 (80%) had PSG scored AHI’s that correlated to the diagnosis of OSA. Eight patients with negative radiographs had AHI consistent with the diagnosis of OSA. The extrapolations from this study are to be taken with caution given such a small sample size, but it appears that lateral radiographs may be a useful tool for screening for the presence of OS A on PSG. However, based on this study, severity cannot be predicted by the size of the adenoids in lateral neck film. Furthermore, given that eight patients with a negative radiograph were also diagnosed with OSA via PSG, the lateral radiograph cannot be used to exclude OSA either. Positioning of the head and whether the lateral neck film was taken while on inspiration or not may also affect the findings, which makes its utility questionable. Again, any two-dimensional static awake imaging may not be representative of dynamic sleep-related airflow obstruction. 

Besides lateral neck radiographs, other imaging modalities such as cephalometry, CT and MR imaging have been reviewed to see if they can replace PSG [39]. These modalities demonstrate to some degree correlation of upper airway imaging and OSA, however most of these studies are more useful in anatomical location and severity of obstruction. They fail to address neuromotor tone or the role of airway collapsibility. There is lack of evidence that any of the above imaging modalities can be used in place of PSG in diagnosing OSA. Cephalometry helps identify areas of possible obstruction utilizing known measurements and ratios, however there are no known studies that look at specificity and sensitivity in diagnosing OSA in pediatric patients. When looking at CT/MR imaging, more emphasis has been put on the MR due to its lack of radiation, but the downside is the need for anesthesia in general for pediatric patients. CT/MRI is performed to determine real-time sites of upper airway obstruction while the child is under anesthesia. MR imaging can be helpful in young infants as there are MR normative values of the upper airway with craniofacial abnormalities [39]. A study was done in 2013 looking at CT imaging and its correlation with AHI in pediatric patients who meet the criteria for OSA. The researchers found a statistically significant correlation between OSA severity, based on AHI, and imaging parameters (*p* = 0.0009), specifically with mean cross-sectional area of the epiglottis to the first thoracic vertebrae [13]. However, this approach cannot be implemented widely as the need for sedation of most children makes it a difficult diagnostic tool. 

### 5.2. Endoscopy

Drug-induced sleep endoscopy (DISE) is performed by using flexible fiberoptic endoscopy and is used to visualize the level of obstruction in the upper airway during sleep. DISE has been increasingly used especially in pediatric patients with residual OSA post adenoidectomy and tonsillectomy (AT). It has also been used prior to AT and during the AT, especially in pediatric patients that are prone to have residual OSA such as in children with Trisomy 21, Prader Willi Syndrome, and craniofacial problems. There have been numerous studies that demonstrate the utility of DISE in the adult population, but the literature in pediatric patients remains scarce and most of the studies involved a small number of patients [40,41]. One area where DISE was postulated to help was in children <2 yrs of age with OSA, since the role of adenoidectomy and/tonsillectomy is poorly defined. The author performed DISE to look at the pattern of upper airway obstruction and to see the value of DISE in therapeutic decision making. The authors noted that in 28 patients 1.5 years or younger with severe OSAS, all but 3 had greater than 50% obstruction at the adenoid level, and 5 had 50% obstruction in the tonsillar area. DISE-directed treatment consisted of adenoidectomy, tonsillectomy, or a combination of both. Results suggested that DISE may be helpful in surgical decision making and that circumferential upper airway narrowing may result in less favorable surgical outcomes [39,40]. There are drawbacks in performing DISE such as cost expense, the need for sedation/general anesthesia, and the need for a highly trained pediatric endoscopist. Recently, there has been some consensus among experts as to when a DISE would be appropriate in children: children with OSA and small tonsils, children with persistent OSA following adenoidectomy and tonsillectomy (T&A), and at the time of AT for children at high risk of persistent OSA [41]. High risk is defined as having severe baseline disease with AHI of 10 events or higher, obesity, craniofacial syndromes including down syndrome and those pediatric patients with neuromuscular disorders [41]. During a pediatric DISE procedure, parameters such as site, pattern, shape, and severity of obstruction should be observed and documented at each anatomic level. The nasal airways, palate/velum, pharyngeal airway, and supraglottic larynx should all be assessed [41].

### 5.3. Biomarkers

A systematic review was published in 2015 which looked at biomarkers to help diagnose OSA in both the adult and pediatric population [42]. OSA and control groups were differentiated by standard PSG protocols and scoring. Nine articles were identified and subjected to qualitative and quantitative analyses. Only one study that was conducted in children and one study conducted in adults that found biomarkers that may be useful for diagnosing OSA. In this study, kallikrein-1, uromodulin, urocortin-3, and orosomucoid-1 were deemed to have enough accuracy to be used as a OSA diagnostic test in children when used in combination with history and physical examination. This study also implied that increased levels of IL-6 and IL-10 in adults may be useful marker in excluding or identifying patients with OSA [42]. Another study done in 2016 by Huang, Y, Gulliemnault, C. et al. showed abnormal levels of HS-CRP, IL-17 and IL-23 in children with OSA [43]. There are limited data on using biomarkers to predict OSA in children and further studies need to be done to validate the utility of such diagnostic tools. These tools, although promising, are far from being able to replace the PSG and cannot yet be used widely for diagnosing OSA.


**Impact**


Pediatric OSA has been linked with behavioral, neurocognitive, and cardiovascular morbidity, as well as metabolic consequences, which mirrors the adult population. By diagnosing children at a young age, we can potentially prevent some of the long-term sequelae of untreated OSA. Children with OSA have been shown to increase healthcare utilization compared to those without OSA [44,45]. Considering this, studies have shown that children with OSA who undergo treatment with adenotonsillectomy do have decreased healthcare cost along with less utilization of healthcare [43,44]. Early, appropriate, and timely diagnosis and treatment has the potential to alter the long-term trajectory of this disease.

## 6. Conclusions

According to the most current literature, PSG has been and remains the gold standard for diagnosing OSA in the pediatric population. There are emerging tools to attempt to hasten the timely diagnosis of pediatric OSA. Most of the tools used as an alternative to PSG are useful for diagnosing severe OSA in children, but their role in otherwise healthy children with mild OSA is yet to be determined.

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
