# Peer review of "Trends in Diagnosing Obstructive Sleep Apnea in Pediatrics"

_children, 2022, doi:10.3390/children9030306_

Round 1
Reviewer 1 Report
Your review is very interesting and useful for pediatrics practice. Really, obstructive sleep apnea (OSA) is a serious problem in the pediatric population, but its definitively diagnosing using in-lab polysomnography (“gold standard”) is associated with limitations related to the attended in-lab polysomnography from limited access to a sleep center, the laborious nature of the test and social/economic barriers, which can delay the diagnosis and treatment. The alternative methods of diagnosis of OSA in children are widely discussed, and can also use if the “gold standard” is not possible, but shouldn’t replace if available. You attempted to highlight current methods of diagnosing OSA in the pediatric population, and compare innovative alternative methods of testing for OSA besides the “gold standard”. Your contribution to this field is needed and valuable for pediatric health care. However, this manuscript can’t accept for publication in the presents form and requires correction:
- In the Abstract and in the main text you should include information sources data.
- In the text, reference numbers should be placed in square brackets, but not in the uppercase.
Author Response
Thank you for reviewing our article. We appreciate all your input. We would like to ask for more clarification to the comment:
"In the Abstract and in the main text you should include information sources data"
we will fix the reference numbers and place them in square brackets.
Reviewer 2 Report
paper is a good review
Author Response
Thank you for your input. We really appreciate your time.